# Immunoinformatics Design and Identification of B-Cell Epitopes from *Vespa affinis* PLA1 Allergen

**DOI:** 10.3390/toxins17080373

**Published:** 2025-07-28

**Authors:** Sophida Sukprasert, Siriporn Nonkhwao, Thitijchaya Thanwiset, Walter Keller, Sakda Daduang

**Affiliations:** 1Chulabhorn International College of Medicine, Thammasat University (Rangsit Campus), Pathum Thani 12120, Thailand; sophida@tu.ac.th (S.S.); siriphorn.nonkhaow@gmail.com (S.N.); thitijchaya_som@hotmail.com (T.T.); 2Protein and Proteomics Research Center for Commercial and Industrial Purposes (ProCCI), Khon Kaen University, Khon Kaen 40002, Thailand; 3Institute of Molecular Biosciences, BioTechMed Graz, University of Graz, 8010 Graz, Austria; walter.keller@uni-graz.at; 4Division of Pharmacognosy and Toxicology, Faculty of Pharmaceutical Sciences, Khon Kaen University, Khon Kaen 40002, Thailand

**Keywords:** phospholipase A1, *Vespa affinis*, linear epitopes, polyclonal antibody, single-chain variable fragment antibody (scFv)

## Abstract

Phospholipase A1 (Ves a 1), a major toxin from *Vespa affinis* venom, poses significant risks to allergic individuals. Nevertheless, the epitope determinants of Ves a 1 have not been characterized. Thus, identifying its linear B-cell epitopes is crucial for understanding envenomation mechanisms. In this study, we predicted and identified B-cell epitopes EP5 and EP6 as potential candidates. EP5 formed an α-helix at the active site of Ves a 1, whereas EP6 adopted an extended loop conformation. Both synthetic peptides were synthesized and evaluated for their inhibitory effects using immune-inhibitory assays with polyclonal antibodies (pAbs) targeting both native (nVes a 1) and recombinant (rVes a 1) forms. The Ves a 1 polyclonal antibodies (pAb-nVes a 1 and pAb-Ves a 1) were produced, and their specificity binding to Ves a 1 was confirmed by Western blot. Next, ELISA inhibition assays showed that EP5 and EP6 significantly blocked pAb binding to both nVes a 1 and rVes a 1. Dot blot and Western blot assays supported these findings, particularly with stronger inhibition toward rVes a 1. Furthermore, enzymatic assays indicated that nVes a 1 and rVes a 1 retained phospholipase activity. Immunoinformatics docking showed that EP5 and EP6 specifically bind to a single-chain variable fragment antibody (scFv) targeting *Naja naja* PLA2. Molecular analysis revealed similar amino acid interactions to the template, suggesting effective paratope–epitope binding. These results support the potential of EP5 and EP6 for future diagnosis and therapy of *V*. *affinis* venom allergy.

## 1. Introduction

Wasp stings cause stent thrombosis, leading to acute myocardial infarction, hemolysis, hepatic dysfunction, oligo-anuria, and azotemia in humans [1]. Vespid venom allergens are known to cause allergic symptom in sensitized individuals. They can induce IgE production that causes type I hypersensitivity, sometimes leading to life-threatening anaphylaxis and allergic asthma [2]. Its major venom components are phospholipase A1 (PLA1), hyaluronidase (Hya), and antigen 5 (Ag5), all of which can cause life-threatening IgE-mediated immunological reactions in humans [3,4]. Among them, only B-cell linear epitopes of Ag5 have been identified and characterized from *Polybia paulista* wasp venom [5]. It has been demonstrated that allergen-specific immunotherapy (AIT) is an effective way of treating the underlying mechanisms of IgE-mediated reactions. The side effects of AIT performed with whole allergen extract have been shown [6]. Therefore, identifying epitopes responsible for allergic responses and designing appropriate hypoallergenic derivatives is crucial for the development of safer AIT. For example, mutations targeting the putative IgE epitope and monomer interface of Equ c 1 reduced dimerization, IgE binding, and allergic response [7].

Vespid venom phospholipase A1s (vPLA1s) are marker allergens commonly used in the diagnosis of Hymenoptera venom allergy. These proteins have 34 kDa in molecular mass and are non-glycosylated, resulting in the absence of carbohydrate cross-reactive determinants (CCDs) typically found in honeybee venom phospholipase A2 (PLA2) [8]. Several vPLA1s have been identified and characterized in different vespid species [9,10,11]. Biological and pharmacological effects of vPLA1s have been reported to promote phospholipid membrane hydrolysis [12], hemolysis [13], inflammation [14] and to activate platelet aggregation leading to thrombosis [15]. After a massive stinging, PLA1 activity triggers several life-threatening pathologies experienced by the victims due to severe hemolysis [16], myocardial dysfunction [13], and cerebral infarction [17]. They are the potent allergenic proteins that cause local inflammatory reactions and type I hypersensitivity [14]. Poly p 1, a vPLA1 from *P*. *paulista*, showed IgE-reactivity by using the sera from ten *P. paulista* venom-allergic patients to rPoly p 1 and nPoly p 1 [18].

Epitopes are surface-exposed regions on the allergen that bind to specific IgE antibodies and elicit a hypersensitivity or allergy. They can be classified into continuous (linear) and discontinuous (conformational) epitopes [19,20]. However, the B-cell-binding epitopes of vPLA1s have not yet been fully studied and characterized. Therefore, identification and characterization of allergenic epitope-binding peptides are essential for the development of advanced component-resolved allergy diagnostics and treatment [21].

Ves a 1, a vPLA1 has been identified previously and characterized as the important major protein among others in *V. affinis* venom. Structural and functional characterization of Ves a 1 helps to understand its role in the envenoming process [22,23]. Additionally, it has also been characterized as a major allergen that shows IgE reactivity with 100% of a panel of wasp allergic patients [24]. However, epitope determinants of Ves a 1 have not yet been identified and characterized. Here, the potential B-cell epitopes of Ves a 1 were delineated by using immune-bioinformatics-based computational tools. Peptides representing a specific B-cell epitope of Ves a 1 were validated by immune-inhibition using polyclonal antibodies produced with native and recombinant forms of Ves a 1.

## 2. Results

### 2.1. Structural Modeling of Ves a 1 and Its B-Cell Epitope Regions Area Analysis

The molecular characteristics of the Ves a 1 model were analyzed by comparing its predicted structure with experimental structures using SWISS-MODEL. The analysis revealed that Ves a 1 shares a 92.98% sequence identity with phospholipase A1 from *Vespa basalis* venom (PDB ID: 4qnn). Ves a 1 is composed of 300 amino acids and features 15 α-helices, 13 β-sheets, and regions of random coils (Figure 1A,B). Linear B-cell epitope predictions were conducted using BepiPred version 2.0, resulting in the identification of seven peptides (EP1–EP7) (Table 1) (Figure 1C,D). Among these, EP5 and EP6 were identified as the most promising potential epitopes, corresponding to the sequences CSHTRAVKYFTECIR and KNPQPVSKCTRNECV, with the highest predicted epitope probabilities of 85.91% and 80.29%, respectively. These peptides were subsequently synthesized and purified, achieving a purity of 95.46% for EP5 and 96.54% for EP6. The molecular masses (MMs) of EP5 and EP6, as determined by ESI-MS, were 1814.12 Da and 1702.97 Da, respectively (Appendix A). These values closely matched their theoretical masses of 1814.11 Da and 1702.96 Da, as calculated using the Compute pI/Mw tool [25].

Additionally, 3D structural models of EP5 and EP6 were generated using AlphaFold, which displayed high confidence scores. The mean predicted local distance difference test (pLDDT) scores were 86.6 and 81.3, and the predicted TM (pTM) scores were 0.0425 and 0.029, respectively (Figure 1E,F). After predicting the 3D structures of the peptides, the results revealed that EP5 predominantly forms an α-helix, while EP6 mainly consists of random coils (Figure 1E,F). Random coils generally exhibit higher antigenicity compared to α-helices and β-sheets, as they are more exposed on the surface and have greater flexibility [26,27]. Furthermore, the prediction of peptide properties indicated the EP6 has the lowest GRAVY score, or the least hydrophobicity, compared to the other peptides, suggesting a high potential to function as a potential antigen (Table 1) [28]. Thus, EP5 and EP6 of Ves a 1 showed different conformations and surface exposure in the context of the folded protein.

### 2.2. Production of the nVes a 1 and rVes a 1-Specific Polyclonal Antibodies

Crude venom (native, n-) and recombinant (r-) of Ves a 1 were separated by SDS-PAGE under reducing condition. As shown in Figure 2A, approximately 25–45 kDa major protein bands were present in nVes a 1 (Lane 1), whereas only one major band with the molecular mass of 53 kDa was observed in rVes a 1 (Lane 2). The proteins with masses of 45, 34, and 25 kDa found in nVes a 1 of *V*. *affinis* venom were identified as hyaluronidase, phospholipase A1, and antigen 5, respectively [22,23]. Specificity and titer determination of anti-sera antibodies to n- and r- of Ves a 1 were determined by using Western blot. As shown in Figure 2, both produced pAb anti-sera showed high titer and were specifically recognized as both n- and r- Ves a 1. The results demonstrated that pAb anti-sera showed strong and specific binding to nVes a 1 (Figure 2B) and rVes a 1 (Figure 2C) with a titer of 1:10,000. The results showed that rVes a 1 was specifically recognized with both pAb anti-sera, suggesting that the two linear epitope determinants, EP5 and EP6, were preserved on the surface of Ves a 1.

### 2.3. Inhibition Assay of Synthetic B-Cell Linear Epitope Peptides

#### 2.3.1. Inhibitory Activity by ELISA

To verify the recognition of Ves a 1 epitopes by polyclonal antibodies, we assessed the inhibitory activities of two synthetic linear peptides, EP5 and EP6, using ELISA assays. Plates were coated with either native Ves a 1 (nVes a 1) or recombinant Ves a 1 (rVes a 1), and antibodies were pre-incubated with each peptide before being added to the wells. As shown in Figure 3A, pre-incubation of pAb-nVes a 1 antibodies with EP5 or EP6 (nVes a 1 + EP5 and nVes a 1 + EP6) significantly reduced antibody binding to nVes a 1 compared to the negative control without inhibitor (nVes a 1, negative) (*p* < 0.05). The positive control, in which pAb-nVes a 1 antibodies were pre-incubated with nVes a 1 (nVes a 1, positive) before being added to the coated wells, also showed significant inhibition. Similarly, Figure 3B shows that pre-incubation with EP5 or EP6 significantly inhibited pAb-nVes a 1 antibody binding to rVes a 1 (rVes a 1 + EP5 and rVes a 1 + EP6), with even stronger inhibition than the positive control (rVes a 1, positive) (*p* < 0.0001). These results indicate that the epitopes represented by EP5 and EP6 are highly conserved between the native and recombinant forms of Ves a 1 and are effectively recognized by pAb-nVes a 1 antibodies.

In contrast, when pAb-rVes a 1 antisera were tested against native Ves a 1 (Figure 3C), neither EP5 nor EP6 caused no significant inhibition (*p* > 0.05), suggesting that pAb-rVes a 1 antibodies have limited recognition of the native protein. However, as shown in Figure 3D, both EP5 and EP6 significantly inhibited pAb-rVes a 1 binding to rVes a 1 (rVes a 1 + EP5 and rVes a 1 + EP6), comparable to the inhibition observed with the positive control (rVes a 1, positive) (*p* < 0.0001). This indicates that EP5 and EP6 epitopes are accessible and specifically recognized by pAb-rVes a 1 antibodies in the recombinant form.

#### 2.3.2. Inhibitory Activity of Synthetic Peptides EP5 and EP6 on the Binding of Polyclonal Antibodies to Native or Recombinant Antigens Evaluated by Dot Blotting and Western Blotting

To further validate the inhibitory activity of synthetic peptides EP5 and EP6, dot blot assays were performed. In Figure 4A, both EP5 and EP6 effectively inhibited the binding of pAb-nVes a 1 antibodies to nVes a 1 (nVes a 1 + EP5, nVes a 1 + EP6 (upper)) and rVes a 1 (rVes a 1 + EP5, rVes a 1 + EP6 (below)), as shown by the reduced spot intensities compared to controls without peptide inhibitors. The positive controls, where antibodies were pre-incubated with their respective antigens (nVes a 1 + nVes a 1 and rVes a 1 + rVes a 1), also showed markedly reduced spot intensities, confirming successful competitive inhibition. In addition, when pAb-rVes a 1 antisera were tested against native Ves a 1 (nVes a 1) in the dot blot assay, pre-incubation with EP5 and EP6 showed only weak inhibitory activity, similar to the limited inhibition observed in the positive control. However, the inhibition intensity measured by ImageJ (version 1.54k) was still higher than that of the control without peptide inhibitors, suggesting that pAb-rVes a 1 antibodies have low but detectable recognition of the native protein [29]. In contrast, when recombinant Ves a 1 (rVes a 1) was tested with pAb-rVes a 1 antisera, pre-incubation with EP5 and EP6 resulted in clear inhibition of antibody binding, with the strongest inhibition observed when antibodies were pre-incubated with rVes a 1 itself (positive control) (Figure 4B; Appendix A). These results indicate that EP5 and EP6 effectively block antibody binding to recombinant Ves a 1 in dot blot assays.

To investigate the specificity of the inhibitory activity related to the structural folding of Ves a 1, Western blot inhibition assays were performed under reducing conditions of SDS-PAGE. As shown in Figure 4C,D, peptides EP5 and EP6 inhibited the binding of pAb-nVes a 1 compared to the control without peptide (nVes a 1, negative), and the control with inhibitor (nVes a 1 + nVes a 1, positive), suggesting that these peptides represent potential B cell-binding epitopes of Ves a 1 specifically recognized by the pAb antisera (Figure 4C). Consistently, EP5 and EP6 also partially inhibited the binding of pAb-rVes a 1, when compared to the control (rVes a 1, negative) and the control with inhibitor (rVes a 1 + rVes a 1, positive) (Figure 4D). Among them, EP6 exhibited slightly stronger inhibitory activity than EP5. These results indicated that the synthetic peptides EP5 and EP6 are potential B cell epitopes of Ves a 1 and may serve as candidate peptides for future vaccine development.

### 2.4. Enzymatic Activity

The phospholipase activity of n- and r-Ves a 1 was examined using a lecithin-plate assay. As shown in Figure 5, cloudy zones were observed with a radius of 24.5 and 15 mm for n- and r- Ves a 1, respectively, when compared to negative controls (5.5 mm). The result distinctly demonstrated that n- and r-Ves a 1 possessed phospholipase activity, cleaving the sn-1 acyl group of phosphatidylcholine and releasing lysophospholipid and free fatty acid.

### 2.5. Molecular Docking

A previous study reported the crystal structure of the scFv antibody targeting *Nja*PLA2C (PDB ID: 8IA6), identifying key antigen-binding residues, including A34, S51, T100, Y165, Y182, H186, and R225 [30]. In this study, alignment analysis revealed that although *Nja*PLA2C shares only 23.89% sequence identity with Ves a 1, several conserved amino acids were present within the antibody-binding site (Figure 6A). Molecular docking analysis of the *Nja*PLA2C-scFv antibody, used as a reference (Figure 6B), which clustered 51 members and achieved the most favorable weighted score of −364.3. However, the docking of Ves a 1 showed that K21 and D71 formed hydrogen bonds with Y165 of the scFv antibody. Additionally, E19 and E239 residues interacted with R225 and Y182, respectively. The docking simulation clustered 48 members and achieved the most favorable result, with a weighted score of −327.1 (Figure 6C). Notably, E239 from the EP5 epitope bound to the Y182 residue within the antibody’s binding site. Furthermore, residues V259, E266, and V268 from EP6 interacted with Y182, Y165, and R225, respectively (Figure 6D,E). Docking simulations for EP5 and EP6 revealed 126 and 169 clustering members, with weighted scores of −237.2 and −262.8, respectively.

## 3. Discussion

Vespid venom phospholipase A1s (vPLA1s), such as that from *V*. *affinis* (Ves a 1), are important allergens responsible for severe allergic reactions and life-threatening systemic envenomation in humans [31,32,33,34]. The biological activities and clinical significance of phospholipase A1 have been well established. Moreover, although a previous study demonstrated that a 34 kDa PLA1 isolated from *V*. *affinis* venom (Ves a 1) exhibited high IgE reactivity in 96% of sensitized patients [35], the epitope determinants of vPLA1s remain largely uncharacterized [5,36]. Therefore, in this study, we identified two novel linear potential B-cell epitopes of Ves a 1 using bioinformatic prediction, synthetic peptide production, and experimental validation.

In this study, we successfully generated polyclonal antibodies specific to native (nVes a 1) and recombinant (rVes a 1) Ves a 1 proteins, with a high titer and strong specificity confirmed by Western blot. The recombinant form of Ves a 1 expressed in *E*. *coli* appeared as a single band of 53 kDa, consistent with its expected molecular weight, while multiple bands were observed in the crude venom, corresponding to hyaluronidase, phospholipase A1, and antigen 5. The structural modeling showed that Ves a 1 shared 92.98% sequence identity with *V*. *basalis* phospholipase A1 (PDB: 4qnn), indicating a highly conserved fold typical of vespid PLA1 enzymes [16,37]. For the identification of potential B-cell epitopes, the epitopes were predicted by BepiPred 2.0, and property analysis highlighted EP5 and EP6 as promising candidates due to their high antigenicity, surface accessibility, and favorable hydrophilicity profiles [38,39]. EP5 adopts an α-helical conformation, while EP6 exhibits a flexible random coil, which may account for its slightly higher antigenicity, as random coil regions are generally more accessible to antibody binding [26,27].

The inhibitory activities of two synthetic linear peptides, EP5 and EP6, were evaluated to map potential B-cell epitopes. ELISA inhibition assays demonstrated that both EP5 and EP6 could significantly inhibit the binding of pAb-nVes a 1 antibodies to both nVes a 1 and rVes a 1, indicating the conservation of epitope determinants between the two forms [40,41]. Furthermore, dot blot analysis confirmed the inhibitory effect of EP5 and EP6, showing reduced spot intensities in the presence of the peptides compared to controls. These results were consistent with those from the ELISA inhibition assays, validating the epitope specificity of the peptides [42,43]. Notably, when pAb-rVes a 1 antisera were tested against nVes a 1, only weak inhibitory activity was observed, indicating possible conformational or structural differences between the native and recombinant forms that affect antibody recognition. Nevertheless, significant inhibition was still observed when pAb-rVes a 1 antisera were tested against rVes a 1 in the presence of EP5 and EP6, confirming that epitope specificity was retained. Importantly, the results demonstrated that both synthetic peptides specifically recognized the paratope-binding sites of pAb, and could obstruct the binding of n- and r-Ves a 1 to pAb antisera. The results suggested that the two potential epitopes may be the main immunogenic determinant domains on the surface of Ves a 1 and spots remain dark. Peptides do not effectively compete with the antigen for antibody binding. In Figure 4B (upper panel), it can be seen that the control inhibitors did not differ significantly from the others, possibly due to greater accessibility or the presence of additional factors facilitating the binding and inhibition of pAb-nVes a 1 and nVes a 1 interactions. If antibody binding is already saturated at the membrane, the addition of more antigen may not effectively compete, resulting in similar signal intensities [44,45].

Interestingly, the inhibitory effect of EP5 and EP6 against pAb-Ves a 1 was observed. This is consistent with the results from Western blot analysis of the linear epitopes of Ves a 1. EP5 and EP6 inhibited the binding of pAb-nVes a 1 and pAb-rVes a 1, suggesting that these peptides may represent potential epitope-binding sites on both native and denatured forms of Ves a 1. Moreover, no cross-reactivity of pAb anti-sera with other proteins was observed, suggesting that pAb anti-sera is specifically recognized only by n- or r-Ves a 1, rendering it a probe detergent to develop vaccine immunotherapy in the future. Therefore, our finding suggests that both peptides may be used as peptide-based vaccines for venom immunotherapy [42]. In addition, the phospholipase activity of both nVes a 1 and rVes a 1 was confirmed through lecithin-plate assays [46]. The results revealed that both Ves a 1 forms retained PLA1 activity, suggestive of correct folding [47]. This reinforces the structural integrity of the recombinant protein and its suitability for use in immunological studies.

Molecular docking simulations provided additional validation of the epitope predictions. Although Ves a 1 shares only 23.89% sequence identity with *Naja naja* PLA2 (*Nja*PLA2C), key residues involved in antibody recognition were conserved. The docking results showed that residues from EP5 (notably E239) and EP6 (V259, E266, V268) formed stable hydrogen bonds with critical antigen-binding residues (Y165, Y182, R225) of a modeled single-chain variable fragment (scFv) antibody based on the *Naja naja* PLA2 crystal structure (PDB: 8IA6) [30,48]. Despite the low sequence identity between Ves a 1 and *Nja*PLA2C, conserved interactions were observed, supporting the relevance of the identified epitopes. Notably, EP6 displayed stronger binding interactions in silico, consistent with its slightly higher inhibition observed experimentally. The structural features of the predicted epitopes of Ves a 1 were highlighted by molecular modeling. The result demonstrated that EP5 and EP6 adopt distinct conformations in the folded protein. Although EP5 contains histidine (H230), which is part of the Ves a 1 catalytic triad (Ser-His-Asp), the synthetic peptide does not impair the enzymatic activity of the native or recombinant protein, as confirmed by the preservation of phospholipase activity [47]. By contrast, EP6 was modeled as a loop area located on the surface of Ves a 1, where it was accessible to the IgG or IgE antibodies. As the finding, these conserved surfaces were proposed to represent the major B-cell-binding epitopes. This agrees with the proposition that coils are mainly found in the region of proteins where surfaces are exposed, making it highly likely that the predicted linear B-cell epitopes are real epitopes [28]. Furthermore, pAb -n and r-Ves a 1 could recognize all structural forms of Ves a 1 to a similar extent. It could suggest that mice pAb antisera may be useful as an allergen detection reagent, and rVes a1 could be developed for allergen immunotherapy of *V*. *affinis* venom allergy. Although docking suggests possible interaction sites, the low sequence identity between Ves a 1 and *Nja*PLA2C limits direct conclusions. Future studies involving site-directed mutagenesis of EP5 and EP6 residues, structural analyses, and antibody-binding assays are necessary to confirm these predicted interactions and to elucidate the precise paratope–epitope recognition mechanisms.

Our findings highlight the potential for developing epitope-specific immunotherapy targeting wasp venom allergens in the future. Furthermore, we demonstrated the feasibility of immunotherapy approaches based on immunoinformatics predictions. Thus, these findings epitope peptides could be used as model peptides to engineer vaccines for allergen immunotherapy and diagnosis of allergies to social wasp venoms.

## 4. Conclusions

Ves a 1 is one established toxin component of Thai wasp, *V*. *affinis* venom. Therefore, identifying epitopes in Ves a 1 is critical for allergen immunotherapy discovery and development. Here, we report predicted potential B cell epitopes in Ves a 1, named EP5 and EP6. They were synthesized and tested for immunoreactivity using specific polyclonal antisera from mice. Both synthetic epitope peptides of Ves a 1 specifically recognized and showed potent inhibitory effects towards native and recombinant forms of Ves a 1 against produced antisera, suggesting that antigenic determinants, in particular B-cell epitopes, often comprise discontinuous epitopes of Ves a 1. Therefore, it may be developed and used as peptides for epitope-specific immunotherapy and for diagnosis of allergies to vespid venoms in the future.

## 5. Materials and Methods

### 5.1. Expression and Purification of Recombinant Ves a 1

The DNA sequencing of Ves a 1 was previously identified by Sukprasert et al., 2013 [22]. Next, the recombinant Ves a 1-pET32a(+) plasmid was transformed into *E*. *coli* BL21(DE3) (Novagen, Madison, WI, USA) to construct the recombinant Ves a 1 protein. Cultures were grown in LB medium at 37 °C to an OD_600_ of 0.6, induced with 0.4 mM IPTG, and incubated for 6 h. Cells were harvested and the pellets were frozen at −80 °C, then washed with washing buffer (0.05 M Tris-HCl pH 8.0, 0.1 M NaCl, 1 mM EDTA) and resuspended in buffer A (50 mM Na_2_HPO_4_ pH 8.0, 300 mM NaCl, 10 mM imidazole, 10% glycerol). The suspensions were sonicated on ice in three 30-sec pulses using a Sonifier^®^ B-30 (output 5, duty cycle 50%). The lysate pellets were resuspended in buffer A with 2 M urea and 1% Triton X-100, followed by sonication under the same conditions. After centrifugation, pellets were washed with buffer A containing 2 M urea and 1% Triton X-100, then with buffer A alone. Finally, proteins were solubilized in buffer A containing 8 M urea and 50 mM DTT by stirring at room temperature for at least 2 h. The suspensions were centrifuged, and supernatants were collected for protein concentration determination and analysis by 13–15% SDS-PAGE under reducing conditions.

After that, the purification was operated with the ÄKTA system (AKTA FPLC system; Amersham Pharmacia Biotech, Uppsala, Sweden) with 1 mL/min flow rate at 4 °C. Buffer A (50 mM Na_2_HPO_4_ (pH 8.0), 300 mM NaCl, 10 mM imidazole, 10% glycerol, 8 M urea, 1 mM DTT and Buffer B (50 mM Na_2_HPO_4_ (pH 8.0), 300 mM NaCl, 250 mM imidazole, 30% glycerol, 8 M urea, 1 mM DTT. Eluted protein fractions were pooled, and TCEP/DTT was added to 15 mM. Urea was reduced stepwise (4 M, 2 M, 1 M urea buffers, each for 4 h) with appropriate NaPP, NaCl, imidazole, glycerol, and DTT concentrations. GSSG was then added to 15 mM, and samples were dialyzed overnight at 4 °C against folding buffer (0.5 M urea, 50 mM NaPP pH 8.0, 75 mM NaCl, 9 mM imidazole, 5% glycerol, 15 mM GSSG). After centrifugation, proteins were analyzed by 13–15% SDS-PAGE and activity assays.

### 5.2. Three-Dimensional Structure Modeling of Ves a 1

The protein sequence of phospholipase A1 from *Vespa affinis* venom (*Vesaf*PLA1) was provided from NCBI database (UniProtKB accession number: P0DMB4). The three-dimensional structure of *Vesaf*PLA1 was modeled by using AlphaFold2, accessible on https://colab.research.google.com/github/sokrypton/ColabFold/blob/main/AlphaFold2.ipynb, accessed on 17 November 2024. The predicted local distance difference test (pLDDT) was used to evaluate the confidence of the models, while the predicted aligned error (PAE) assessed the positional accuracy of each amino acid. Modeling was performed with three recycles and an RMSD tolerance of 0.5 Å. PLDDT, ranked single-chain models and complex structures were ranked using the predicted TM-score. The top five models from each run were selected, and their pTM and pLDDT scores were recorded [49]. The protein structures were then visualized using PyMOL (version 2.5.0).

### 5.3. B-Cell Epitope Determinant Prediction and Peptide Synthesis

B-cell linear epitope determinants of Ves a 1s were identified by using immunobioinformatics-based computational tools [28], DNAStar Protean system (http://www.dnastar.com/t-protean.aspx, accessed on 21 June 2020), Bioinformatics Predicted Antigenic Peptides website (http://nrs.harvard.edu/urn-3:HUL.InstRepos, accessed on 30 June 2020), and BepiPred 2.0 Server (http://www.cbs.dtu.dk/services/BepiPred/, accessed on 30 June 2020) [47]. Moreover, the linear epitopes were calculated the properties by using NovoPro Peptide Property Calculator (https://www.novoprolabs.com/tools/calc_peptide_property, accessed on 30 June 2020). Next, peptides corresponding to the regions that were predicted as potential epitopic sites were synthesized at 1st BASE, Malaysia. The purity of the peptides was greater than 95% as assessed by high-performance liquid chromatography (HPLC). The molecular mass of the synthesized peptides was confirmed by Electrospray Ionization–Mass Spectrometry (ESI-MS). Theoretical pI and Mw of linear epitope peptides were computed by the Compute pI/Mw tool (http://web.expasy.org/compute_pi/, accessed on 1 July 2020). The peptides were stored at −20 °C until used. The structures of the linear B-cell epitopes were predicted by using the AlphaFold2, as previously described [50].

### 5.4. One-Dimensional Sodium Dodecyl Sulfate–Polyacrylamide Gel Electrophoresis (SDS-PAGE)

Before starting the SDS-PAGE procedure, the protein contents were quantitatively determined by the Bradford method using bovine serum albumin (BSA) as the standard [51]. SDS-PAGE was performed under reducing conditions using 13–15% (***w*/*v***) acrylamide resolving gel and 4% (***w*/*v***) acrylamide stacking gel, following standard protocols with β-mercaptoethanol and DTT (dithiothreitol) added to the sample buffer. Phosphorylase B (97 kDa), BSA (66 kDa), chicken ovalbumin (45 kDa), carbonic anhydrase (30 kDa), trypsin inhibitor (20.1 kDa), and α-lactalbumin (14.4 kDa) were used as standards. After the samples were applied to the gel, the proteins were resolved at 150 V for 1 h. The gels were then stained with Coomassie staining solution.

### 5.5. Mouse Immunizations

The animal experiment was approved by Thammasat Animal Care and Use Committee no. 007/2557. After electrophoresis, purified toxin from crude venom and recombinant protein responding to Ves a 1 band were excised from the gels and frozen at −80 °C. Mouse polyclonal antibodies specific to nVes a 1 (pAb-nVes a 1) and rVes a 1 (pAb-rVes a 1) were produced [52]. Briefly, male ICR mice aged 8 weeks were immunized intraperitoneal with swollen Ves a 1 and rVes a 1 bands in PBS buffer (135 mM NaCl, 1.5 mM KH_2_PO_4_, 2.5 mM KCl and 8.0 mM Na_2_HPO_4_, pH 7.4) mixed with Freund’s complete adjuvant (Sigma-Aldrich, St. Louis, MO, USA) in an antigen to adjuvant ratio of 1:1. Anesthetized mice were immunized 2-week intervals with the emulsion prepared from the gel suspension and Freund’s incomplete adjuvant. After the ninth or tenth booster, blood was collected by trimming from the intravenous vein of tail and maintained at 4 °C after collection. The serum was centrifuged at 10,000× *g* at 4 °C for 10 min, and then the supernatant containing the antiserum was pooled and used for further studies.

### 5.6. Evaluation of Ves a 1 Antigenicity by ELISA

Crude venom protein and recombinant Ves a 1 were diluted to 20 μg/mL in carbonate buffer, pH 9.5. The ELISA was carried out in a 96-well plate (Nalge Nunc International, Denmark). Each well of a 96-well polystyrene plate was coated with 50 μL of 20 μg/mL diluted venom (as antigen) and incubated at 4 °C overnight. The unbound antigen was washed with PBS, pH 7.4, containing 0.05% (***v***/***v***) Tween-20. The wells were blocked with 100 μL blocking solution (5% (***w***/***v***) skimmed milk in PBST) for 1 h at 37 °C. After three rinses with PBST, 1:1000 ***v***/***v*** of the serum was diluted in blocking solution and then incubated in each well for 1 h at 37 °C. After subsequently washing with PBST, the plate was incubated with 50 μL of conjugated anti-mouse IgG linked with alkaline phosphatase at 37 °C for 1 h. After washing with PBST again, the plate was further washed with PBS three times. Bound antiserum was detected using freshly prepared chromogenic substrate (1 mg/mL of ρ-nitrophenyl phosphate, 100 mM Tris-HCl, pH 9.5, 100 mM NaCl, and 50 mM MgCl_2_). The absorbance was measured at 405 nm using an ELISA microplate reader (Varioskan FLASH, Thermo Scientific, Waltham, MA, USA) [53].

### 5.7. Immunoblot Inhibition Assay by ELISA, Dot Blot and Western Blot

ELISA inhibition assays with the peptides were performed in 96-well plates using the same protocol as described in the previous section. The microplate was coated with crude venom (nVes a 1) or recombinant Ves a 1 (rVes a 1), and was then blocked with 5% defatted milk. EP5 and EP6 peptides were dissolved in PBS buffer (135 mM NaCl, 1.5 mM KH_2_PO_4_, 2.5 mM KCl and 8.0 mM Na_2_HPO_4_, pH 7.4), and diluted to a final concentration of 1 µg/mL for use in the assay. Further, the antisera antibodies (pAb-nVes a 1 and pAb-rVes a 1) were pre-incubated with peptides EP5 or EP6 before being added to the coated wells. The remaining steps were the same as the ELISA. Inhibition was presented as an absorbance decrease in the sample with peptides. Controls included wells exposed to antibodies without any peptides. Optical density (OD) values obtained for a mice pAb binding with no peptides, crude or rVes a 1 were normalized to 100% binding. The percentage of mice inhibition on pAb binding was calculated as follows:$$Inhibition\;\left(\%\right)=100\times \;\frac{OD\;\left(M-\right)-OD(M+)}{OD\;(M-)}$$

M+ and M− antibodies with and without peptides or crude venom or recombinant proteins, respectively; OD, optical density. Ves a 1 itself was used as the inhibitor.

For dot blot analysis, 1 µg of crude venom or recombinant proteins was dotted in duplicates on nitrocellulose membranes (Schleicher & Schuell, Dassel, Germany). Membranes were then exposed to 1:1000 anti-sera antibodies, which were pre-absorbed with either crude venom or recombinant or synthetic peptides EP5 and EP6 at 4 °C overnight. The AP-conjugate substrate kit detected bound antiserum (Bio-Rad, Hercules, CA, USA). Dot blot intensity was quantified using ImageJ (version 1.54k) by setting the mean gray value as the measurement parameter [29,54]. For Western blot, after electrophoresis, the gel was placed into a blotting apparatus, and the proteins were transferred to a nitrocellulose membrane for 1 h. The membrane was incubated with blocking solution (5% nonfat dry milk in PBST buffer). It was also incubated with anti-serum diluted in blocking solution and with goat anti-mouse IgG linked with alkaline phosphatase for 1 h at room temperature. The blotted protein bands were detected by using an AP-conjugate substrate kit (Bio-Rad, Hercules, CA, USA) [55].

### 5.8. Enzymatic Activity Assay

Phospholipase activity of protein samples was performed by using modified lecithin-plate assay as previously described [20]. Briefly, PCY mixture (1% agarose gel, 0.8% soybean lecithin, 1% NaCl, 0.25% taurocholic acid, and 20 mM CaCl_2_) was prepared on Petri dishes. Samples diluted in 50 mM Tris-HCl, pH 8.0, were well loaded. After incubation for 24 h, halo sizes were observed.

### 5.9. Molecular Docking of Ves a 1 and Epitopes with scFv Antibody

The sequences of Ves a 1 and phospholipase A2 from *Naja naja* venom (GenBank: AAA66029.1) were retrieved from the NCBI database and aligned using MEGA11 software, employing Clustal Omega for sequence alignment (https://www.ebi.ac.uk/jdispatcher/msa/clustalo?stype=protein, accessed on 19 November 2024). Conserved amino acids were identified with AlignmentViewer 2.0 (http://www.alignmentviewer.org/, accessed on 20 November 2024). Molecular docking was used to analyze the binding energies and types of intermolecular interactions between Ves a 1, EP5, and EP6 with the paratope specific binding site of scFv antibody (PDB ID: 8IA6). Moreover, the crystal structure of scFv antibody against phospholipase A2 of Echis carinatus venom complex was applied as a template (PDB ID: 8IA6) sourced from RCSB PDB (https://www.rcsb.org/, accessed on 17 November 2024) [30,48]. Molecular docking of *Vesaf*PLA1 and epitopes with scFv antibody were conducted using ClusPro 2.0, a highly regarded protein–protein docking server (https://cluspro.org/help.php, accessed on 30 November 2024). The PIPER protein interaction energy was computed using the equation *E* = *w_1_E_rep_* + *w_2_E_attr_* + *w_3_E_elec_* + *w_4_E_DARS_*, where *E_rep_* and *E_attr_* denote the repulsive and attractive components of van der Waals interaction energy, *E_elec_* represents the electrostatic energy term, and *E_DARS_* reflects the pairwise structure-based potential derived using the “decoys as the reference state” (DARS) method [56]. Protein–protein complexes interacting were visualized using the PyMOL program.

### 5.10. Statistical Analysis

Data were reported as mean ± standard error of mean (SEM). Significant differences were determined by using one-way analysis of variance (ANOVA) followed by Dunnett’s post hoc test using GraphPad Prism 5 (GraphPad software Inc., San Diego, CA, USA). *p* value ≤ 0.05 was set as the significance threshold.

## Figures and Tables

**Figure 1 toxins-17-00373-f001:**
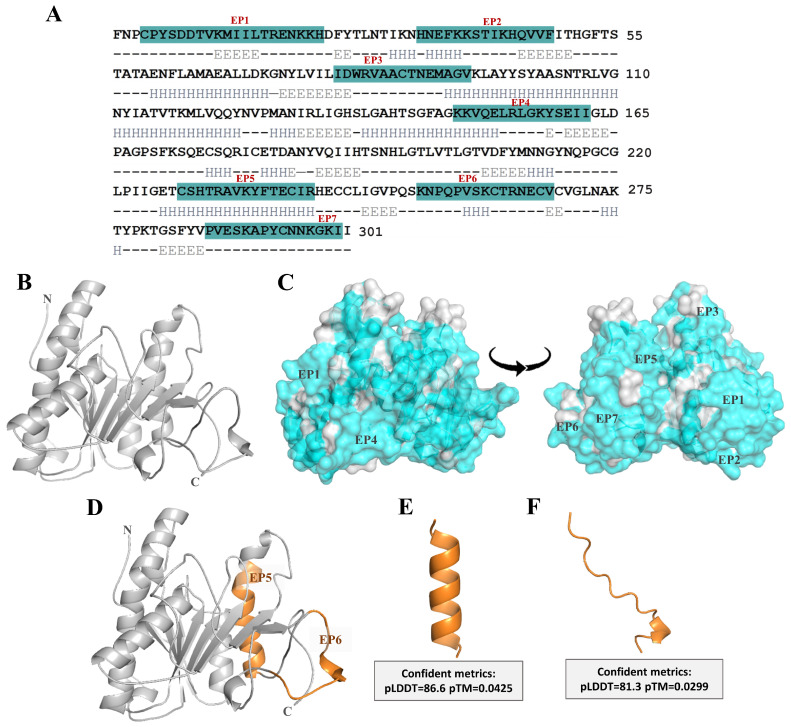
(**A**) The deduced amino acid sequence and electronic address of UniProtKB P0DMB4. Blue labels represent B-cell epitopes 1–7 on Ves a 1 areas. (**B**) The structural features of Ves a 1 were modeled by using Alphafold prediction. (**C**) The surface mapping of conformation B-cell epitopes (EP1–EP7). Gray and blue surfaces indicate hydrophobic and hydrophilic regions, respectively. (**D**–**F**) The three-dimensional structures of EP5 and EP6 after generated by Alphafold.

**Figure 2 toxins-17-00373-f002:**
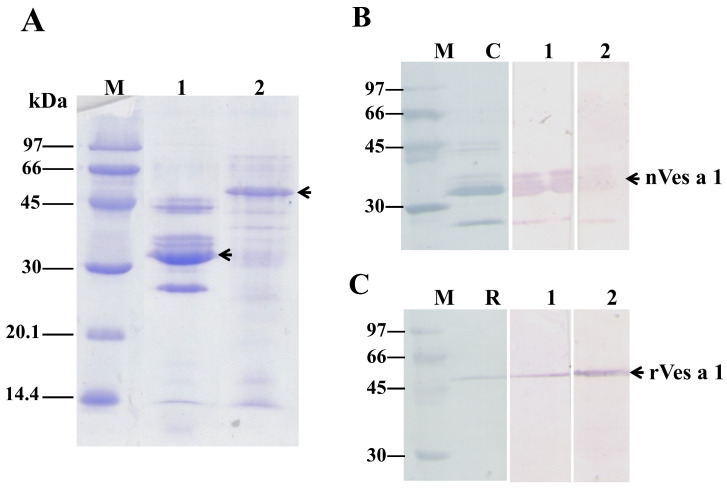
(**A**) 15% SDS-PAGE analysis of nVes a 1 (native) and rVes a1 (recombinant). M, low molecular mass marker (GE Healthcare, Sweden); lane 1, crude venom protein; lane 2, recombinant protein of Ves a 1 expressed in *E*. *coli*. Black arrows corresponded to n- and r-Ves a 1 protein bands were indicated. Specificity and titer determination of mice pAb antisera by Western blot. Crude venom protein (**B**) and recombinant protein (**C**) were transferred onto nitrocellulose membrane. M, low molecular mass marker; C, crude venom protein; and R, recombinant protein were stained with 0.1% amido black. Lane 1 and 2 were probed with pAb-nVes a 1 and pAb-rVes a 1 antisera, respectively. An antibody titer of 1:10,000 was used for all.

**Figure 3 toxins-17-00373-f003:**
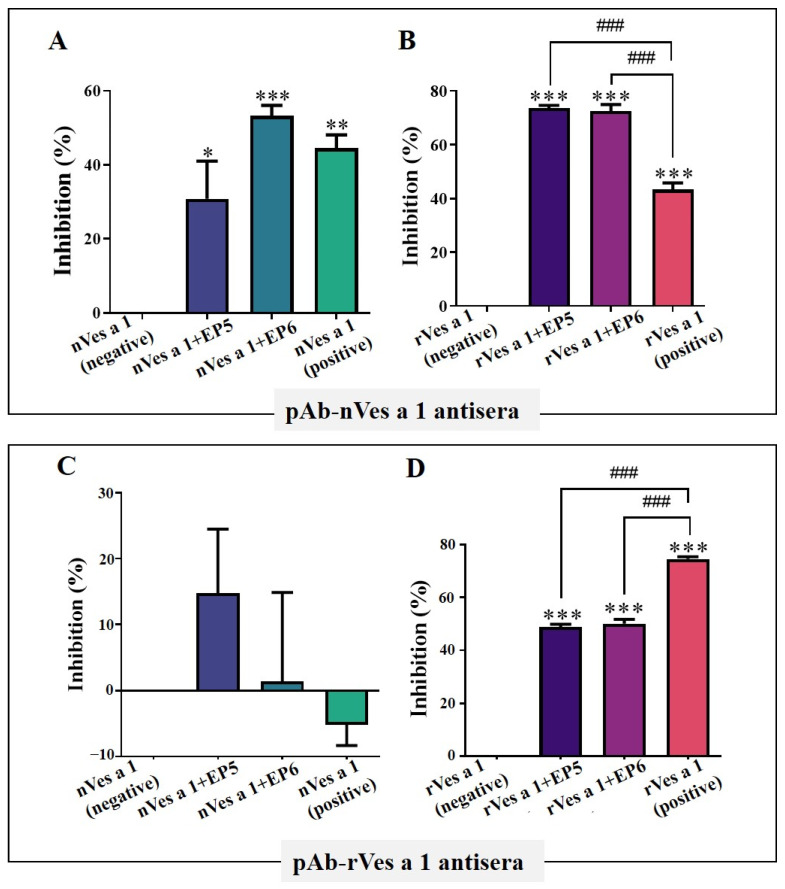
Inhibitory activity of synthetic peptides EP5 and EP6 determined by ELISA using pAb-nVes a 1 antisera (**A**,**B**) and pAb-rVes a 1 antisera (**C**,**D**). Native crude venom protein (nVes a 1) and recombinant Ves a 1 (rVes a 1) were coated on the plates. Wells coated with antigen alone and incubated without antisera served as negative controls. Wells coated with nVes a 1 or rVes a 1 and incubated with their respective antisera were designated as nVes a 1 (positive) and rVes a 1 (positive), respectively. Data are presented as mean ± SEM (*n* = 3). Significant differences are indicated by * for *p* < 0.05, ** for *p* < 0.001, *** for *p* < 0.0001, and ### for *p* < 0.0001. The symbols * and # denote comparisons of negative and positive control inhibitor, respectively.

**Figure 4 toxins-17-00373-f004:**
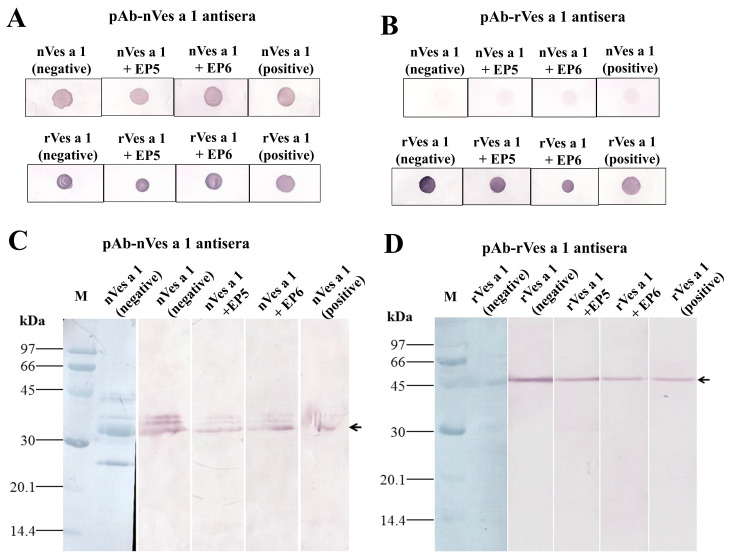
Inhibitory activity of synthetic peptides EP5 and EP6 determined by dot blot and Western blot assays. Dot blot analysis using pAb-nVes a 1 antisera (**A**) and pAb-rVes a 1 antisera (**B**). Native crude venom protein (nVes a 1 (upper)) and recombinant Ves a 1 (rVes a 1 (below)) were dotted onto nitrocellulose membranes. Antisera were pre-incubated overnight at 4 °C with either EP5 or EP6 as the respective antigens before probing. Antisera were used at a dilution of 1:1000. For the reducing condition, Western blot analysis was used, showing the specificity and inhibitory activity of EP5 and EP6 against pAb-nVes a 1 antisera (**C**) and pAb-rVes a 1 antisera (**D**). Proteins were separated by SDS-PAGE, transferred onto nitrocellulose membranes, and probed with antisera pre-incubated overnight at 4 °C with the peptide (EP5 and EP6). M indicates the low-molecular-weight protein marker. Lanes nVes a 1 and rVes a 1 represent incubation without peptide (negative controls). Pre-incubation with nVes a 1 and rVes a 1 served as positive inhibition controls. Black arrows indicate the positions of native and recombinant Ves a 1 proteins.

**Figure 5 toxins-17-00373-f005:**
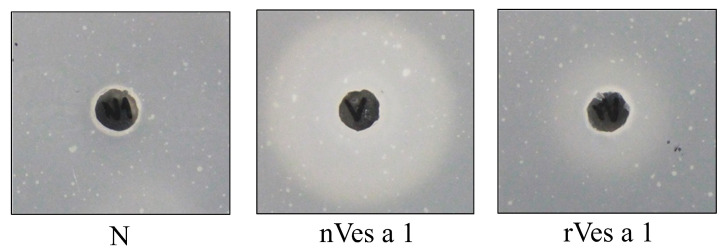
PCY-lecithin-plate assay of n- and r- Ves a 1. Twenty micrograms of crude venom and recombinant Ves a 1 were mixed with buffer before loading into the well. Buffer N1 (50 mM Tris-HCl, pH 8.0) was used as a negative control. The reaction mixture was loaded and then incubated at room temperature for 24 h. After incubation, halo sizes were measured.

**Figure 6 toxins-17-00373-f006:**
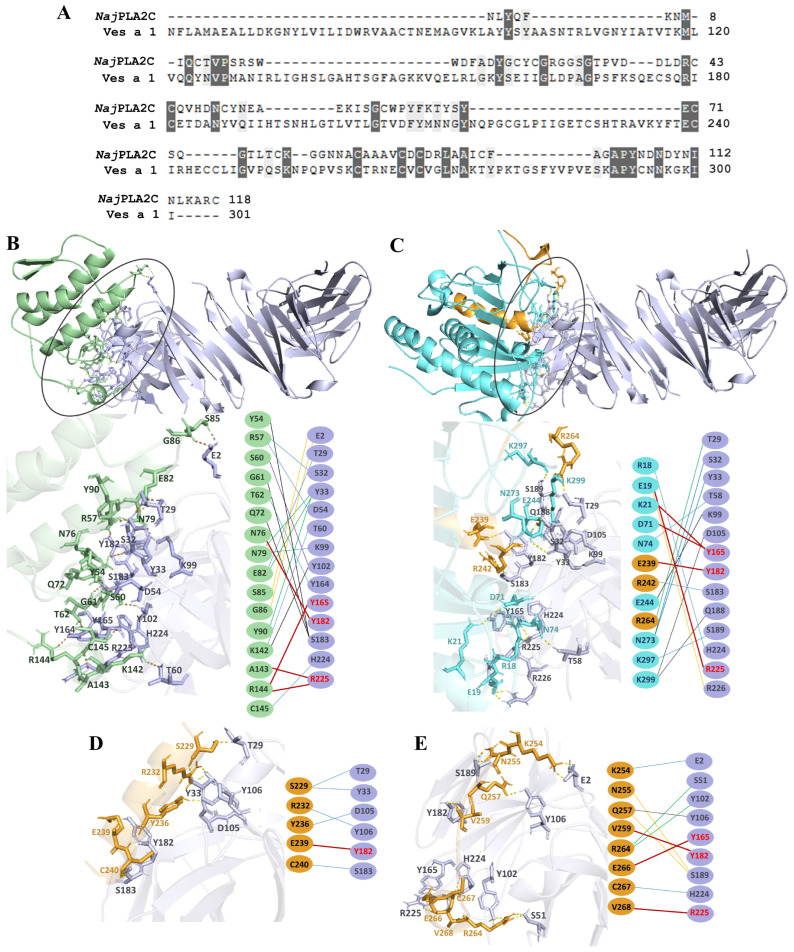
(**A**) Amino-acid-sequence alignment of Ves a 1 (UniProtKB: P0DMB4) and *Naj*-PLA2C phospholipase A2 from *Naja naja* venom (GenBank: AAA66029.1). Dark and light gray labeled letters indicate amino acid identity and similarity, respectively. (**B**,**C**) The molecular docking of *Naj*PLA2C (green residue) and Ves a 1 (cyan residue) complex with scFv antibody (purple residue) from crystallographic against phospholipase A2 of *Echis carinatus* venom (PDB ID: 8IA6). (**D**,**E**) The interaction of EP5 and EP6 with scFv antibody, respectively. Red letters indicate the scFv antibody’s antigen-binding site. The yellow dotted line represents H-bonding.

**Table 1 toxins-17-00373-t001:** Identification of linear B-cell epitope sequences of Ves a 1 from V. affinis venom.

Epitope	Sequence	Position	Length (Residue)	Probability Scale ^a^	Net Charge	GRAVY ^b^
EP1	CPYSDDTVKMIILTRENKKH	4–23	20	70.00	1.10	−0.86
EP2	HNEFKKSTIKHQVVF	34–48	15	69.19	2.20	−0.77
EP3	IDWRVAACTNEMAGV	80–94	15	69.34	−1.00	0.38
EP4	KKVQELRLGKYSEII	148–16	15	76.73	2.00	−0.56
EP5	CSHTRAVKYFTECIR	228–242	15	85.91	1.82	−0.32
EP6	KNPQPVSKCTRNECV	254–268	15	80.29	1.90	−1.17
EP7	PVESKAPYCNNKGKII	286–301	16	65.70	1.90	−0.65

^a^ Probability scale in percentage (0–100%) for each epitope (values above 60% indicate potential epitopes). ^b^ The Grand Average of Hydropathicity (GRAVY) measures the hydrophobicity of a polypeptide by averaging the hydropathy values of its amino acids, using the Kyte–Doolittle scale. Higher positive scores indicate greater hydrophobicity.

## Data Availability

The original contributions presented in this study are included in the article. Further inquiries can be directed to the corresponding author.

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
