# Peer review of "Immunoinformatics Design and Identification of B-Cell Epitopes from Vespa affinis PLA1 Allergen"

_toxins, 2025, doi:10.3390/toxins17080373_

Round 1

Reviewer 1 Report

Comments and Suggestions for Authors

Reference:  “Immunoinformatics Design and Identification of B-Cell Epitopes from Vespa affinisPLA1 Allergen”. Manuscript submitted to Toxins June 2025.

General comments:  In this manuscript the authors are working with the magnificent toxin Phospholipase A1 found in wasp venoms, and in this case in the venom of the wasp Vespa affinis. Due to the lack of knowledge described in the literature about the allergenic epitopes found in this toxin, the authors are studying these challenges. Throughout the work, the authors describe the identification of two linear epitopes called EP5 and EP6 with potential activities. The first epitope (EP5) comprises an alpha helix secondary structure and is part of the enzyme's active site, and the second (EP6) comprises a loop structure. The authors produced synthetic peptides and polyclonal antibodies that recognize venom phospholipase A1 and recombinant isoform, and analyzed the reagents obtained through competition and inhibition experiments using ELISA, WB and Dot B. According to the authors the data found indicate that the peptides were able to inhibit the binding of polyclonal antibodies to wild-type or recombinant toxins. Additionally, trough bioinformatics analyses, authors demonstrated the potential binding of the epitopes found in antibodies that recognize PLA2 from Naja naja venom. The authors discuss the potential of reagents produced in advances in knowledge in the area to improve therapy and diagnosis of accident victims. After reading the text, I believe it fits within the scope of TOXINS. Its main subject is the analysis of allergenic toxins PLA1 in venoms and wasps. However, the text needs to be refined before it can be accepted for TOXINS. In my opinion, the manuscript has confusing text, the legends need to be rewritten, the figures also need to be redone, especially the Dot Blotting inhibitions, the expression of the recombinant toxin shows a difference in mass in relation to the wild-type toxin, which needs to be explained, the authors do not provide details of the expression of the recombinant toxin, the method of obtaining antiserum using bands eluted from SDS-PAGE has criticism, and finally the docking data are speculative and the recombinant toxin isoforms need to be expressed to proven hypothesis. Enclosed I suggest changes that could improve the quality of manuscript.  

Specific Comments:

1- In the line 37  ….  anti-gen 5 (Ag5). Please change to antigen 5, the right form to write. 

2- In the line 42 ….Side effects of AIT performed with whole allergen extract have been shown.  Please indicate references for this sentence. 

3- In the lines 43 and 44 …the design of appropriate hypoallergenic derivatives is crucial for the development of safer AIT. Here authors could indicate some examples of hypoallergenic derivatives, as the term is ample.  

4- In the lines 46 and 47 …These proteins are approximately 34 kDa in size. Better change to …  These proteins have 34 kDa in molecular mass … 

5-   Among lines 46 to 50    ….These proteins are approximately 34 kDa in size and are non-glycosylated, resulting in the absence of carbohydrate cross-reactive determinants (CCDs) typically found in honeybee venom phospholipase A2 (PLA2). In my opinion some references could be incorporated in this part of text.  

6-  Lines 50 and 51…. vPLA1s have been reported to promote membrane hydrolysis [10].  Better change … to promote phospholipid membrane hydrolysis.. 

7- Lines 71 and 72 … Peptides representing specific B-cell epitope peptides of Ves a 1 were validated… Please remove the second word peptide in the revised text … Peptides representing specific B-cell epitope of Ves a 1 were validated…  

8- In the line 77 …The characteristics of the Ves a 1 model. In the revised text change to ….The molecular characteristics of Ves a 1 …. 

9- In the lines 85 and 86 … with the highest probabilities of 85.91 % and 80.29 %, respectively. But highest probability of what? Authors need to explain . 

10- In the lines 89 and 90… closely matching the theoretical values of 1814.11 Da and 1702.96 Da (data not shown). If there is a published article about this, then a citation would be interesting.  

11- In the lines 102 and 103 … which may explain the different inhibitory effects towards the pAb antisera.  This sentence, although important, is out of context since the authors have not yet presented their inhibition experiments. Transfer this conclusion to the correct place, after the inhibition experiments. 

12- In the line 104, … Table1. Identification of linear B-cell epitope sequences of Ves a 1 from V. affinis venom. In the revised version of manuscript I would incorporate in the tittle   … by using    BepiPred version 2.0 Program.  

13- About figure 1 legend. …. (A) Shows the deduced amino acid of Ves a 1 (UniProtKB: P0DMB4).  Please put the word ….amino acid sequence and electronic address of UniProtKB. 

14- Still in figure 1A the authors could identify by numbers the different epitopes shown. 

15- In figure 1C authors must inform what means different colors? They are positive and negative charges and hydrophobic regions?  

16- In the line 120 … with the molecular size of 53 kDa. In the revised text please change molecular size to molecular mass.  

17- In the lines 120 and 121 … The proteins of size 45, 34, and 25 kDa found in… Please change to   … the proteins with mass of ….kDa  . 

18- In the legend of figure 2 , lines 135 and 139 …molecular weight… In the revised text change to molecular mass. Molecules have mass. Although literature accepts the term, according a scientific rigor the right term is molecular mass. There is Mass Spectrometry, but not Weight Spectrometry. 

19- In the line 135 … low molecular weight marker  . Change to… Low molecular mass markers. 

20- In the line 136 … lane 2, recombinant protein of Ves a 1 expressed in E. coli. Following scientific rigor, the authors should indicate the strain of E. coli used and the purification method, in addition to commenting that there was an enrichment in the recombinant protein content, but not complete purification, and finally describe the method used to express and purify the recombinant toxin, construction, details…. 

21- Still on figure 2A, some comment should be made about the differences in masses observed for the native protein with a lower molecular mass and the recombinant protein with a higher molecular mass. What is the explanation? 

22- In column 1 of Figure 2B, there is a failure in the WB reaction. I assume that the authors have repeated the experiment, do they not have a better reaction that shows positivity for all antigen band and substitute the one shown? 

23- Also, in the column 2 of figure 2B, there is a low positivity when authors used polyclonal antibodies against recombinant toxin in the reaction upon crude venom. Any explanation for this low reactivity? 

24- There is a band below 30 kDa pointed in lanes C, 1 and 2, with cross reacted with antibodies. Any comment of authors about this? Is this band a fragment of antigen? 

 25- Finally, …the title of figura 2 could be changed … 2.2. Production of the nVes a 1 and rVes a 1-specific polyclonal antibodies and titer determination. In my opinion, the authors did not determine the titration of the serum obtained, this presupposes other experiments. In my oppinion in the revised text a better title could be …. “Production of the nVes a 1 and rVes a 1-specific polyclonal antibodies” 

26- About figure 3 and legend. Figure 3A: The text in the description of the results, the legend of figure 3 and the experiments are confusing. 

27- The data actually show inhibition in the ELISA reaction, compared to the negative control, but they seem confusing to me. 

28- In my opinion, the authors should sensitize the plate with the native toxin, or the recombinant toxin alone and incubate with the respective polyclonal sera. This would be the 100% positive reaction. Change the letters C by nVes a 1 and R by rVes a 1.  

29- The inhibition reactions would be performed using specific sera pre-incubated with the respective toxins alone and then the mixture would be evaluated in the ELISA with the sensitized toxins on the plate. This should be the control of greatest inhibition, since these were the antigens used to obtain the respective sera. 

30- The sera would then be pre-incubated with the studied synthetic peptides P5 and P6 alone, and then tested on the ELISA plate against the sensitized toxins. It should give a partial inhibition, since there are other different epitopes in the recombinant and recombinant toxins, as shown in Figure 2.  What is the reason for the authors mixture peptides and native or recombinant toxins during inhibition reactions?  The competition to have value should have been made using the peptides alone. 

31- I would like to hear the opinions of other reviewers, but the text in figure 3 is confusing. It needs to be redone. 

32-  About figure 4, item 2.3.2 … 2.3.2. Inhibitory activity by Dot blot and Western blot. The sentence is confusing and incomplete. The authors need to describe exactly what was done! 

33- In the revised text write…Inhibitory activity of synthetic peptides EP5 and EP6 on the binding of polyclonal antibodies to native or recombinant antigens evaluated by Dot Blotting and Western Blotting.   

34- In the legend of figure 4, line 198, it is not clear whether the authors used crude venom or purified toxin from the venom. As it is written crude venom protein, it could be crude venom instead of toxin purified from crude venom. 

35- If the authors are using the purified toxin from the venom then it is better to write: Wild-type toxin purified from the crude venom was used …. 

36- Don't make legends for letters A and B together. Make legends for the topics separately. So making one legend for Letter A and another for Letter B! It gets confusing and it's not the Reviewer's job to do that. 

37-  In lines 174 and 175 the authors wrote  …. a dot blot inhibition assay was performed, and spot intensities were quantified using ImageJ [26]. But the quantifications were not shown in the text. Any reason for that? 

38- Although there is no explanation in the legend and text, I imagine that the top panel of Fig. 4A represents the wild type toxin incubated without competition. Therefore, there is no reason for the authors to write the abbreviations of the peptides at the top. Nor to repeat several Dots. I imagine that this would be the positive control of antigen-antibody binding. This figure could be called fig. 4A-1 

39- In the upper panel of Fig. 4B, I imagine that it is a competition between peptides and polyclonal serum against wild-type toxin. This is where inhibition is observed. But the authors need to make this clear. Put numbers in Fig. 4B-1.  

40- In Figure 4A below, which could be called Fig. 4A-2, it appears to be the reaction of the recombinant toxin with its respective serum. Positive reaction. Also in this case there was no incubation with the respective peptides. Would this be a positive control? In this case there is no need to place the peptides in the figure, since there was no competition. Is this what I understand? 

41- Finally, Fig. 4B below, which could be called Fig. 4B-2, would be the reaction with competition between the studied peptides and respective serum or serum against wild-type toxin? Is confused. In this case, in my point of view that was no inhibition. 

42- In summary, the data shown by DoT Blottings are confusing and show inhibition only for the competition between peptides, wild-type toxin and respective serum. The data using recombinant toxin are confusing. 

43- That difference in mass between the wild-type toxin and the recombinant toxin, shown in Fig. 2, leaves these experiments in suspense. Why does the recombinant toxin have such a large mass difference compared to the wild-type? It is indeed the same toxin. Did the authors perform some minimal control to ensure expression? A mass spectrometry to ensure that they are the same proteins? An amino-terminal sequencing? 

44- The competition results in WB are also confusing. The authors should show other results, since the data shows positivity in half of the image. As can be seen in Fig. 4C.  

45- The data shown in Fig. 4D are more realistic, with clearer and more coherent images. There was indeed competition between the peptides and the serum tested. In this case, the explanation would be that in WB the antigens are denatured, where linear epitopes prevail, which are more similar to synthetic peptides, which are linear. 

46- The enzymatic activity experiments shown in Fig. 5 make it clear that the recombinant toxin obtained by the authors has its problems, since its activity is very low compared to the wild-type toxin. 

47- The text written between lines 209 and 215 is incomplete. What do N1 and N2 mean? Why did the authors use different buffers in the incubation reactions? Just the buffer in which the toxins were incubated would be sufficient. The authors need to explain the text better. 

48- Were different buffers used with the two toxins? This needs to be explained in the text. And why? 

49- Why did the authors perform experiments on enzymatic activity? Isn't the focus of the manuscript to study the immunological aspects of antigens? The ideal would be to perform experiments on inhibition of enzymatic activity by sera. 

50- About the text where the authors discuss the data on molecular docking and figure 6, lines  223 a 228   …. A previous study reported the crystal structure of the scFv antibody targeting NjaPLA2C (PDB ID: 8IA6), identifying key antigen-binding residues, including A34, S51, T100, Y165, Y182, H186, and R225 [27]. In this study, alignment analysis revealed that although NjaPLA2C shares only 23.89 % sequence identity with Ves a 1, several conserved amino ac-ids were present within the antibody-binding site (Figure 6A). 

 51- Is it reliable to compare the structural data of a phospholipase A1 with the data of a phospholipase A2? Especially if the amino acid residues identity between them is only 23.89%? 

52- In my opinion the data shown in Figure 6 are speculative, especially figs. B, C, D and E and would need to be confirmed with mutated isoforms at the discussed amino acid residues for the data to be reliable. 

53- In the line 373 change … calcu-lated  by calculated  . 

54- In the line 385 … determined by the Bradford method . Please indicate a reference for text.

 55- In the lines 386 and 387 … SDS-PAGE was performed following the standard method using 13-15% (w/v) resolving gel and a 4% (w/v) stacking gel. Do the percentages indicated refer to acrylamide concentrations? Is yes then complete the text. Also indicate if gel was performed under reduced or not-reduced conditions? Refine methodology. 

56- The production of polyclonal sera against wild-type and recombinant toxins was performed using bands cut from SDS-PAGE gels. This is not a very good criterion since the method performs one-dimensional separation and any other protein with the same mass will be mixed together. Alternatively, the authors could purify the antigens by other methods. 

57- Line 407 … Crude venom protein. Please substitute by…. Purified Toxin from crude venom …

58- Topic 5.5, lines  406 to 421. 5.5. Titer determination by Enzyme-linked Immunosorbent Assay (ELISA) and Western blot. What the authors did was not a titration of the serum obtained, but rather a binding reaction of the antibodies to the antigen fixed on the plate. Titration is something else. See Antibodies: A Laboratory Manual  Edward Harlow, David Lane CSHL Press, 1988 for details.   

59- Although recombinant toxin it is a key reagent in the studies, the authors do not describe obtaining the recombinant toxin in M/M. 

60- In the lines 432 and 433 …. Further, it was incubated with pre-incubated mixture of anti-sera antibodies. This sentence is incomplete. Pre-incubated with what? Authors need to detail their procedures in M/M! 

61- Lines 445 and 446 … Dot blot intensity was quantified using ImageJ by setting the mean gray value as the measurement parameter [26]. These data, although discussed, were not shown in the manuscript! 

Author Response

Thank you for your suggestions. That significantly improves our work.

For the comments and responses file has been attached here.

Reviewer 2 Report

Comments and Suggestions for Authors

The study was well-developed, and the results corroborated the in silico prediction. Despite more sophisticated techniques for protein-protein interaction being available today, the methods used were effective in testing the research hypothesis.
No additional comments or criticisms. 

Author Response

We appreciate your valuable time spent reviewing our work, which has helped us improve it.

Reviewer 3 Report

Comments and Suggestions for Authors

The experiments are well planned, executed, and described. I have no major comments on these parts of the manuscript. However, I found a few shortcomings that should be corrected/clarified before the manuscript is published. A detailed list is provided below. 

l 85 NH2-...sequence   index “2” should be in superscript. 
However, it would be best to refrain from showing the N and C ends (NH2 and COOH). They are not included in Table 1. Currently, there are two (identical sequences) in the manuscript, shown in slightly different ways. I would recommend standardization by removing the terminal functional groups.

l 89-90 The molecular masses calculated and determined by ESI MS are almost identical. However, ESI MS directly generates the mass for the pseudomolecular ion M+H (possibly multiply charged), which is 1 greater than MW. Why is this effect not visible in the results presented?

l 373, 381 remove hyphen

l354 The Methods section lacks information on how (in which buffer, at what pH, at what concentration) the peptide solution(s) was/were prepared.

l429 Section 5.6   There is no information on the quantities of peptides ep5 and 6 that were used.

Abbreviation section

EP5  .... peptide not peptides

EP6 the same comment

In the main text and in the Abbreviations section, different symbols are used for PLA1 (PLA1 vs PLA1-in superscript). The same applies to PLA2.
Two different symbols are also used to denote the molecular weight of peptides (MM-l88 and Mw-379). 

Author Response

We appreciate your valuable time spent reviewing our work, which has helped us improve it.

The experiments are well planned, executed, and described. I have no major comments on these parts of the manuscript. However, I found a few shortcomings that should be corrected/clarified before the manuscript is published. A detailed list is provided below.

Comments: l 85 NH2-...sequence   index “2” should be in superscript.

However, it would be best to refrain from showing the N and C ends (NH2 and COOH). They are not included in Table 1. Currently, there are two (identical sequences) in the manuscript, shown in slightly different ways. I would recommend standardization by removing the terminal functional groups.

Response: Thank you so much for your observation. We have standardized this point, and removed the NH2 and COOH from the Line 85.

Comments: l 89-90 The molecular masses calculated and determined by ESI MS are almost identical. However, ESI MS directly generates the mass for the pseudomolecular ion M+H (possibly multiply charged), which is 1 greater than MW. Why is this effect not visible in the results presented?

Response: We have added more data support to the supplementary file. Additionally, we have deleted the phrase (“data not shown”), added the appropriate citation, and revised the sentence accordingly (Line: 102-103).

The molecular masses (MM) of EP5 and EP6, as determined by ESI-MS, were 1814.12 Da and 1702.97 Da, respectively (Figure S1A‒B). These values closely matched their theoretical masses of 1814.11 Da and 1702.96 Da, as calculated using the Compute pI/Mw tool.

Comments: l 373, 381 remove hyphen

Response: The hyphen has been removed.

Comments: l354 The Methods section lacks information on how (in which buffer, at what pH, at what concentration) the peptide solution(s) was/were prepared.

Response: Thank you so much for your suggestion.

We added the buffer and its concentration that was used to dissolve the peptides.

In the main text Line (481-484):

EP5 and EP6 peptides were dissolved in 50 mM carbonate–bicarbonate buffer, pH 9.5, and diluted to a final concentration of 1 µg/mL for use in the assay. Further, the antisera antibodies (pAb-nVes a 1 and pAb-rVes a 1) were pre-incubated with peptides EP5 or EP6 be-fore being added to the coated wells.

Comments: l429 Section 5.6   There is no information on the quantities of peptides ep5 and 6 that were used.

Response:  We appreciate your comment. According to a previous comment, we added the details of the buffer and the peptides concentration in the main text already.

Comments: Abbreviation section

EP5  .... peptide not peptides

EP6 the same comment

Response: Thank you for your observation. We have revised this point.

Comments: In the main text and in the Abbreviations section, different symbols are used for PLA1 (PLA1 vs PLA1-in superscript). The same applies to PLA2.

Two different symbols are also used to denote the molecular weight of peptides (MM-l88 and Mw-379).

Response: Thank you for your comment.

We have changed the superscript formatting of PLA₁ and PLA₂ in the abbreviations to plain text (PLA1 and PLA2) to match the formatting used in the main text.

Round 2

Reviewer 1 Report

Comments and Suggestions for Authors

After reading the response letter sent by the authors, I believe that this revised version is improved over the first version. The authors were polite and courteous in their responses, demonstrating willingness and knowledge of the field. Although I continue to find criticisms in the manuscript, in this revised version, in my opinion, and if this the opinions of the other reviewers and the Editorial Board are the same, the text has the potential to be approved for publication by TOXINS.